# Shortening the Biliopancreatic Limb Length of One Anastomosis Gastric Bypass Maintains Glucose Homeostasis Improvement with Limited Weight Loss

**DOI:** 10.3390/jcm11174976

**Published:** 2022-08-24

**Authors:** Lara Ribeiro-Parenti, Hounayda El Jindi, Alexandra Willemetz, Matthieu Siebert, Nathalie Kapel, Johanne Le Beyec, André Bado, Maude Le Gall

**Affiliations:** 1UMR-S 1149 Centre de Recherche sur l’Inflammation, Université Paris Cité, Inserm, 75018 Paris, France; lara.ribeiro@aphp.fr (L.R.-P.); hounayda.shahrour@inserm.fr (H.E.J.); alexandra.willemetz@inserm.fr (A.W.); matthieu.siebert@aphp.fr (M.S.); johanne.le-beyec@inserm.fr (J.L.B.); andre.bado@inserm.fr (A.B.); 2Service de Chirurgie Digestive œsogastrique et Bariatrique, Hôpital Bichat—Claude-Bernard, Assistance Publique-Hôpitaux de Paris, Université Paris Cité, 75018 Paris, France; 3Department of Coprology, Assistance Publique-Hôpitaux de Paris, Hôpital Pitié-Salpêtrière-Charles Foix, Sorbonne Université, 75013 Paris, France; nathalie.kapel@aphp.fr; 4Service de Biochimie Endocrinienne et Oncologique, Hôpital Pitié-Salpêtrière-Charles Foix, Assistance Publique-Hôpitaux de Paris, Sorbonne Université, 75013 Paris, France

**Keywords:** bariatric surgery, one anastomosis gastric bypass, mini bypass, glucose homeostasis

## Abstract

One anastomosis gastric bypass (OAGB) is associated with similar metabolic improvements and weight loss as Roux-en-Y gastric bypass (RYGB). However, this bariatric procedure is still controversial as it is suspected to result in undernutrition. Reducing the size of the biliopancreatic limb of OAGB could be essential to maintain positive outcomes while preventing side effects. The objective of this study was to compare and contrast outcomes of OAGB with two different biliopancreatic limb lengths to RYGB and Sham surgery in obese and non-obese rats. Lean and diet-induced obese Wistar rats were operated on RYGB, OAGB with a short (15 cm OAGB-15) or a long (35 cm OAGB-35) biliopancreatic limb or Sham surgery. Body weight and food intake were monitored over 30 weeks, and rats underwent oral glucose and insulin tolerance tests with a pancreatic and gut hormone secretion assay. Macronutrient absorption was determined by fecal analyses. Statistical analyses used non-parametric one-way or two-way ANOVA tests. Compared to Sham rats, RYGB, OAGB-15 and OAGB-35 rats displayed a significant reduced weight. Weight loss was greater after OAGB-35 than after OAGB-15 or Sham surgery because of transient malabsorption. All OAGB- and RYGB-operated rats displayed an improved pancreatic and gut hormone secretion in response to a meal compared to Sham rats, these effects were independent of limb length, rat weight, and maintained overtime. In conclusion, glucose homeostasis was similarly improved in obese and non-obese OAGB-15 and OAGB-35 rats suggesting that shortening the biliopancreatic limb can improve the metabolic parameters without a major influence on weight.

## 1. Introduction

Although standard primary treatment of obesity consists of diet and lifestyle interventions, in the absence of long-term success, bariatric surgery is still currently the only effective treatment for managing morbid obesity, as it leads to rapid and long-lasting weight loss. Bariatric surgery also allows an improvement or a cure of certain obesity-associated disorders such as type 2 diabetes.

All bariatric surgeries are a remodeling of the gastrointestinal tract. The Roux-en-Y Gastric Bypass (RYGB) was initially performed to induce food restriction through the creation of a 20 to 50 mL gastric pouch (in humans) and nutrient malabsorption bypassing the rest of the stomach and the proximal part of the small intestine. This surgery creates three intestinal segments; a 150 cm alimentary limb (AL), which receives undigested food, a 50 to 75 cm biliopancreatic limb (BPL), which receives secretions from the stomach, gall bladder and pancreas, and finally a common limb (CL), which makes the connection between the two previous limbs. This operation has proven to be very effective in the long-term on weight-loss and resolution of comorbidities, but it also presents chronic complications including strictures, internal hernias, gastro-gastric fistulae, dumping syndrome, and nutritional deficiencies [1,2].

Surgeons are always trying to find better procedures with the best results and less invasiveness for patients, involving simpler surgeries, shorter operating times and hospital stays. As a result, in 1997, Rutledge developed a new procedure, the One Anastomosis Gastric Bypass (OAGB), also called the mini-bypass or omega loop gastric bypass [3]. In its original description, OAGB consisted of placing a long, narrow gastric tube with a gastro-jejunal anastomosis 200 cm from the Treitz angle. This procedure thus avoids the jejunojejunostomy of RYGB while achieving significant weight loss [3] and remission of diabetes [4]. OAGB is a rising procedure because it is simpler and associated with outcomes comparable to RYGB in terms of weight loss and comorbidities improvement [5]. However, despite the promising results, OAGB remains controversial [6], as it could be responsible for functional complications such as chronic diarrhea and steatorrhea leading to undernutrition [4], gastric acid and bile reflux causing gastroesophageal reflux disease (GERD), anastomotic ulcers, and at risk of esophageal cancer [7]. In most performed OAGB, the BPL length is 200 cm [8], but some authors suggest that reducing the length to 150 cm may reduce the risk of nutritional complications without decreasing efficacy [9,10,11].

The aim of this study was to assay whether reducing the length of the BPL in OAGB could lead to similar positive outcomes on glucose control in obese and non-obese rats compared with RYGB and OAGB with a long BPL.

## 2. Materials and Methods

### 2.1. Animals

All animal studies comply with the ARRIVE guidelines. They were conducted in accordance with EU directive 2010/63/EU for animal experiments and approved by the Institutional Animal Care and Use Committee (CEA N° 121) and the French Ministry of Higher Education and Research (APAFIS #8290). Rats were housed in a conventional animal facility, with a 12 h day/12 h night cycle.

Six-week-old Wistar male rats (n = 24) (Janvier, Le Genest St Isle, France) were fed an ad libitum high-fat diet with 45% lipids (Altromin—Special diet C1090-45 Genestil, Royaucourt, France) for four months before undergoing bariatric surgery. When they weighed between 568 and 786 g (mean ± SEM 666 g ± 11), they were randomly assigned to Sham surgery (n = 6), RYGB (n = 6), OAGB-15 (n = 6) or OAGB-35 (n = 6). One OAGB-15 animal died 9 days after the surgery, and one OAGB-35 animal died 1 day after the surgery. In addition, after the cull, we noticed that one OAGB-35 animal had a defective gastric pouch, this animal was thus excluded from the study. The final number of animals was n = 6 for Sham, n = 6 for RYGB, n = 5 for OAGB-15, and n = 4 for OAGB-35.

A second set of 44 six-week-old Wistar male rats (Janvier, Le Genest St Isle, France) were fed ad libitum normal chow diet (ND), and when they weighed between 255 and 341 g (mean ± SEM 302 ± 4), they were randomly assigned to the non-operated group (n = 8), RYGB (n = 12) OAGB-15 (n = 12) or OAGB-35 (n = 12). One RYGB, one OAGB-15 and two OAGB-35 died during or shortly after the surgery. The final number of animals was n = 8 for control, n = 10 for RYGB, n = 11 for OAGB-15, and n = 10 for OAGB-35.

### 2.2. Surgery

Detailed surgery procedures were already described [12,13]. Briefly, in rats operated on RYGB, the jejunum was transected 15 cm distally from the duodenojejunal angle. The Roux limb was anastomosed to the gastric pouch, and the biliopancreatic limb was anastomosed 20 cm distal to the gastro-jejunal anastomosis. As the total length of a rat bowel is about 100 cm, this would be equivalent to a BPL of 85 cm and an AL of 120 cm in humans. For rats operated on OAGB, the jejunum was anastomosed to the gastric pouch 15 cm (OAGB-15) or 35 cm (OAGB-35) from the duodenojejunal angle. As the total rat jejunum is around 100 cm long, the 35 cm in rats would be equivalent to the commonly realized 200 cm BPL in humans, whereas the 15 cm would be closer to a less than 100 cm BPL. For Sham rats, to mimic surgery, the stomach was pinched with an unarmed staple gun similar to the one used to perform the gastric pouch in RYGB and OAGB animals [12,13], and the jejunum was unrolled.

For post-operative care, rats were kept in individual cages and monitored every 12 h for 48 h and then daily for weight, food intake, general behavior and wound healing with administration of analgesics and antibiotics (buprenorphine 0.05 mg/kg, meloxicam 1 mg/kg, penicillin 20,000 units/kg), if necessary. All rats (including control groups) were kept with no access to water and food during day 1 post-surgery, but hydration was assured through injection of an isotonic polyionic solution (2 g KCl/L, 4 g NaCl/L, 50 g glucose/L). On day 2 and 3 post-surgery, rats had free access to water and a liquid highly digestible diet composed of 21% protein and 79% carbohydrate (Altromin C0200, Genestil, Royaucourt, France), and on day 4, they had access ad libitum to a normal chow diet.

### 2.3. Intestinal Nutrient Absorption

Macronutrient absorption was determined by fecal analyses as previously described [14]. Briefly, rats were kept in metabolic cages from postoperative day 45 to 49. The total amount of food consumed was recorded, and stools were collected daily for 3 days. The total energy content in food and stools was determined by bomb calorimetry (C200 bomb calorimeter, IKA-Werke GmbH & Co., Staufen im Breisgau, Germany). Intestinal energy absorption represented the percentage of ingested calories actually absorbed by the intestine (ingested calories—calories lost in fecal outputs)/ingested calories X100.

### 2.4. Meal Test, Oral Glucose Tolerance Test and Insulin Tolerance Test

For pancreatic and gastrointestinal hormone secretion, rats were fasted for 16 h before being subjected to an oral gavage of glucose (1 g/kg body weight) and a combination of sunflower, rapeseed, and grape seed oil (Isio4, Lessieur 1 mL/kg body weight). Blood was sampled from the tail vein before (T0) and 20 min after the gavage, and collected in the presence of heparin and DPPIV inhibitor (Roche, Indianapolis, IN) to prevent degradation of active GLP1. Rat plasma concentrations of active GLP1, PYY, GIP, Insulin, C peptide and leptin, were quantified on a Luminex MagPix200 analyzer using a Milliplex rat gut hormone panel (RMHMAG-84 K; Merck Millipore, Guyancourt, France).

Rats were fasted for 16 h before being subjected to an oral glucose tolerance test (OGTT) and 5 h before being subjected to an insulin tolerance test (ITT). Blood was sampled from the tail vein before (T0) and 5, 15, 30, 60, 90, and 120 min after the oral load of glucose (1 g/kg body weight) for OGTT or the intraperitoneal injection of insulin (0.5 U/kg body weight) for ITT. Blood glucose levels were measured with the AccuChek System (Roche Diagnostics, Meylan, France) and expressed in mg·dL^−1^.

### 2.5. Statistical Analyses

Statistical analyses were performed using GraphPad Prism 7.04 (GraphPad, La Jolla, CA, USA). Kruskal–Wallis and Dunn’s post hoc tests were used to determine significant differences between groups. Repeated-measures (RM) analyses of variance with a Dunnett’s multiple comparison tests were used for analysis of weight and glycaemia over time. 

All statistical results were considered significant when *p* < 0.05.

## 3. Results

### 3.1. Shortening the BPL in Obese Rats Affects Body Weight but Not Food Intake and Absorption in the Long Term

As previously observed [15], in obese rats, all bypass surgeries induced a rapid weight loss, and maximal weight loss was observed between 10 days (for OAGB) and 18 days (for RYGB) after surgery (Figure 1A). Whereas the Sham and OAGB-15 animals experienced around 10% of weight loss compared to their preoperative weight (−7.1 ± 0.4% for Sham vs. −12.4 ± 0.9% for OAGB-15, *p* = 0.24), the OAGB-35 and RYGB groups experienced the greatest weight loss 15 days post-operations (OAGB-35 −17.4 ± 1.3% and RYGB −18.3 ± 2.5% *p* < 0.01 vs. Sham).

After 2 weeks, all animals operated on bariatric surgery regained some weight, and after 40 days (6 weeks) they all stabilized their weight up to the end of the experiment. By that time, the Sham rats had almost returned to their preoperative body weight (−4.5 ± 1.2%), whereas the weight of the OAGB-15, OAGB-35 and RYGB rats stabilized at −8.7%, −11.8% and −10.0% less than their preoperative weight, respectively (Figure 1A). The differences in body weight were paralleled with the reduced plasma leptin levels measured in the fasted state 75 days (11 weeks) post-surgery (Figure 1B), reflecting the reduced adipose tissue mass.

After stabilization of body weight, we assayed nutrient absorption and glucose control in Sham and operated animals. During the 6th week post-surgery, food intake and stool excretion were slightly increased in OAGB-35 and RYGB, but the overall intestinal absorbed calories (expressed as percentage of food intake) were not different between groups despite a trend toward reduction in OAGB-35 (Figure 1C–E). 

### 3.2. Shortening the BPL in OAGB Does Not Affect Enterohormone Secretion in Obese Animals

To assay glucose regulation and enterohormone secretion, rats were given a test meal (composed of lipids and sugars) 11 weeks after surgery and glycaemia, insulin, C-peptide, but also gut-produced GLP1, GIP and PYY were assayed before and 20 min after the gavage (Figure 2). 

In the fasted state, plasma glucose was not statistically different between groups. Twenty minutes after the oral load of meal, glycaemia was increased by 1.5 to 1.7-fold and there was no significant difference between the groups.

Whereas C-peptide and insulin were not statistically different between groups in the fasted state or 20 min after the gavage, stimulation of their secretion increased after bariatric surgery compared to Sham animals. For instance, insulin secretion was increased 5.6-fold after OAGB-15, 6.2-fold after OAGB-35 and 4.4 fold after RYGB versus only 2.7 fold in Sham animals (*p* = 0.025 all bariatric surgery groups together versus Sham).

In all groups, GIP levels were induced by the meal (Figure 2D). However, compared to the Sham, the meal-induced secretion of GIP was higher in RYGB animals, but not in OAGB-15 or OAGB-35 rats.

As expected, GLP1 secretion was strongly increased in animals undergoing bariatric surgery compared to Sham animals (Figure 2E). Whereas meal-induced secretion of GLP1 was aborted in obese Sham animals, it was restored after bariatric surgery (X 3.9 after OAGB-15, X 2.9 after OAGB-35 and X 3.6 after RYGB versus X 1.3 after Sham *p* < 0.05) without any difference within the different procedure groups. More specifically, no difference was observed in OAGB versus RYGB rats, and the limb length of the OAGB had no effect on the altered GLP1 secretion. 

Fasted and stimulated levels of PYY were also modified after bariatric surgery (Figure 2F). Not only PYY secretion in response to the meal was increased, but basal PYY levels were decreased after bariatric surgery resulting in a 10-fold induction of secretion in operated animals compared to Sham animals (*p* = 0.046). 

To complete the study of glucose regulation, we performed an oral glucose tolerance test (OGTT) on fasted obese rats 8 weeks post-surgery (Figure 3A) and an insulin tolerance test (ITT) 10 weeks post-surgery (Figure 3B).

After the glucose gavage, OAGB- and RYGB-operated rats presented a similar hyperglycemic peak at 30 min compared with Sham animals. However, after 30 min, the restoration to basal glycaemia was more rapid in RYGB and OAGB-35 animals than in Sham and OAGB-15 animals (Figure 3A). The glycaemia response to intraperitoneal injection of insulin was identical in Sham and OAGB-15 rats, but insulin sensitivity was more important in OAGB-35 and RYGB animals (Figure 3B). Accordingly, calculations of HOMA IR and Fasted Glucose insulin ratio tended to decrease, whereas insulin secretion index tended to increase in OAGB-35 and RYGB animals compared to OAGB-15 and Sham animals without reaching statistical significance (Figure 3C–E). This set of results demonstrates that the restoration of insulin sensitivity after bariatric surgery in this model is more related to weight loss than enterohormone secretion.

As reducing the BPL length in OAGB did not significantly affect enterohormone secretion compared to a longer BPL despite a difference in weight loss, we hypothesis that reducing the length could maintain a positive effect on glucose metabolism while preventing risk of undernutrition. It also suggests that using a short BPL could be interesting to treat moderately obese diabetic or prediabetic subjects. To test this hypothesis, we reproduced some of our experiments in non-obese rats.

### 3.3. OAGB with a Short BPL Ameliorates Glucose Regulation in Non-Obese Animals

We analyzed the evolution of glucose tolerance and insulin sensitivity overtime in non-obese animals operated on OAGB or RYGB. Detailed body weight loss and food intake of these animals were previously reported [15].

As shown in Figure 4 inserts, 10 and 25 weeks post-surgery, body weight was reduced in all animals operated on with bariatric surgery in comparison to non-operated animals, but this reduction was not significantly different between OAGB-15, OAGB-35 and RYGB. 

After 10 weeks and compared to control animals, both OAGB- and RYGB-operated rats presented a higher hyperglycemic peak 15 min after a glucose gavage in agreement with previous reports in obese individuals. However, after 30 min, the restoration to basal glycaemia was more rapid in animals operated on bariatric surgery than in non-operated animals, suggesting that glycemic control is actually improved after bariatric surgery consistent with the data presented for obese animals (Figure 1).

Interestingly, whereas non-operated animals displayed a degraded glucose tolerance over time (10 weeks vs. 25 weeks, Figure 4A vs. Figure 4B), with a higher early peak and a later return to normal as they get older and bigger, animals operated on bariatric surgery maintained a similar response over time despite gaining weight. These results suggest that the impact of bariatric surgery on glucose tolerance was not solely dependent of weight, persisted over time and did not depend neither on the procedure nor on the length of the BPL. It should be noted that all operated animals displayed a better insulin sensitivity compared to non-operated animals 28 weeks after surgery, as expected from their reduced weight (Figure 4C).

## 4. Discussion

This study demonstrates that reducing the BPL length in OAGB resulted in a difference in weight loss, but did not affect enterohormone secretion compared to a longer BPL, suggesting that reducing the length of the BPL could maintain a positive effect on glucose metabolism. It also suggests that using a short BPL could be interesting to treat moderately obese diabetic subjects as glucose homeostasis was similarly improved in non-obese animals.

In this study, we report that shortening the BPL affects body weight, but not food intake and absorption in the long term. These data are not in contrast with our previous reports of an important malabsorption in the early days post-surgery in OAGB-35 animals [14,15], but indicate that malabsorption in OAGB-35 rats is transient, probably due to adaptive processes (i.e., increased mucosa surface) of the remodeled gastrointestinal tract after bariatric surgery [12,14,16].

In this model of OAGB with a short BPL and despite a slight increase in biliary reflux in the upper gastro-intestinal tract, a reduced incidence of eso-gastric lesions was reported compared to OAGB with a long BPL [15]. In addition, shortening the BPL reduced fecal loss during the early weeks post-surgery and the risk of undernutrition [15]. Accordingly, a recent study in humans, analyzing the conversion of Sleeve Gastrectomy to OAGB using two lengths of BPL (150 cm vs. 200 cm) reported fewer long-term nutritional and functional complications in patients with 150 cm BPL compared with 200 cm BPL [11]. Altogether, these results strongly suggest that reducing the length of the BPL could be a pertinent strategy to reduce the risk of undernutrition while maintaining the positive impact on glucose homeostasis without increasing the bilio-pancreatic reflux consequences in the long term.

Compared to the Sham animals, the meal-induced secretion of GIP was higher in RYGB animals, but not in OAGB-15 or OAGB-35 rats. Difference in GIP response to bariatric surgery has already been reported in humans, and it was suggested that the length of excluded limb—where a major number of GIP cells are located—could explained this effect [17]. Our results do not support this explanation, but suggest that the RYGB procedure has a specific effect compared to the OAGB procedure independent of the BPL length. On the contrary, after all types of bariatric surgery, GLP1 and PYY secretion in response to the meal were increased, and in addition, basal PYY levels were decreased. GLP1- and PYY-expressing cells are essentially located in the distal part of the gastrointestinal tract, thus our results comfort the hypothesis that rapid delivery of partly digested food in the distal part of the gut increases the stimulation of distal enteroendocrine cells expressing PYY and GLP1 [18]. Of note, the restoration of insulin sensitivity after bariatric surgery in this rat model is more related to weight loss than enterohormone secretion.

The small number of subjects in each group constitutes an important limitation of our study preventing us from multiple comparison analyses. Still, our findings support the hypothesis that, by decreasing the length of the BPL, it is possible to achieve a satisfactory glucose control in obese and non-obese animals. The validation of our observations in humans will require a randomized controlled trial comparing enterohormone secretions in OAGB with a BPL < 150 cm versus OAGB with a BPL 200 cm and RYGB. 

In conclusion, our results demonstrate that shortening the BPL of OAGB allows similar positive outcomes on enterohormone secretion and glucose metabolism in obese and non-obese rats, suggesting this strategy could be interesting as a metabolic surgery for moderately obese diabetic subjects.

## Figures and Tables

**Figure 1 jcm-11-04976-f001:**
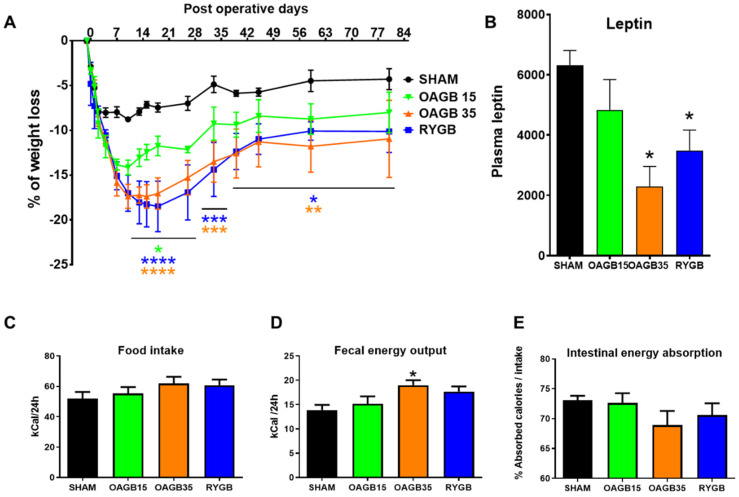
**Shortening the BPL of OAGB affects body weight, but not food intake and absorption.** (**A**) Weight loss and regain according to surgical procedure during 24 post-operative weeks expressed as % of initial weight. * *p* < 0.05; ** *p* < 0.01, *** *p* < 0.001, **** *p* < 0.0001 for OAGB-15 (green) OAGB-35 (orange) or RYGB (blue) vs. Sham (black) animals by Dunnett’s multiple comparison tests after 2-way RM ANOVA. (**B**) Plasma leptin levels in fasted animals 75 days post-operation. * *p* < 0.05 versus Sham animals by Dunn’s multiple comparison tests after Kruskal–Wallis Test. (**C**–**E**) Daily food intake (**C**), fecal energy outputs (**D**) and calculated intestinal energy absorption (**E**) during the 6th post-operative week. * *p* < 0.05 for OAGB-35 vs. Sham, no statistical difference was observed after Kruskal–Wallis Test. Data are expressed as means ± SEM, SHAM n = 6, RYGB n = 6, OAGB15 n = 5, OAGB35 n = 4.

**Figure 2 jcm-11-04976-f002:**
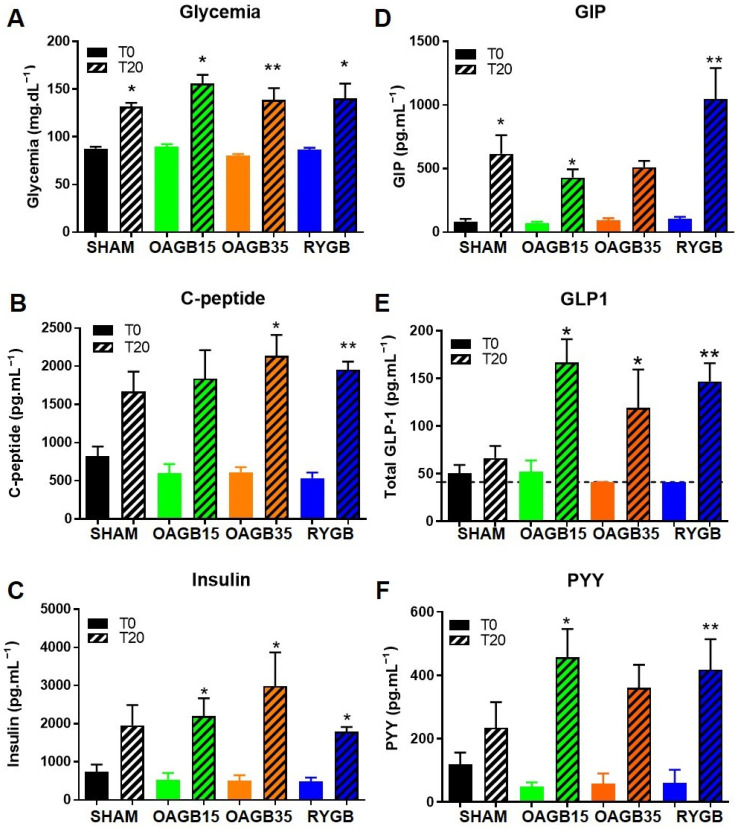
**Shortening the BPL of OAGB preserves the altered secretion of pancreatic and gut hormones in response to a meal**. Glycaemia (**A**) C-peptide (**B**), insulin (**C**), GIP (**D**), GLP1 (**E**) and PYY (**F**) levels in the plasma of rats before or 20 min after oral administration of a meal composed of glucose and lipids. * *p* < 0.05; ** *p* < 0.01, T20 versus T0 by Dunn’s multiple comparison tests after Kruskal–Wallis Test. Data are expressed as means ± SEM, SHAM n = 6, RYGB n = 6, OAGB15 n = 5, and OAGB35 n = 4.

**Figure 3 jcm-11-04976-f003:**
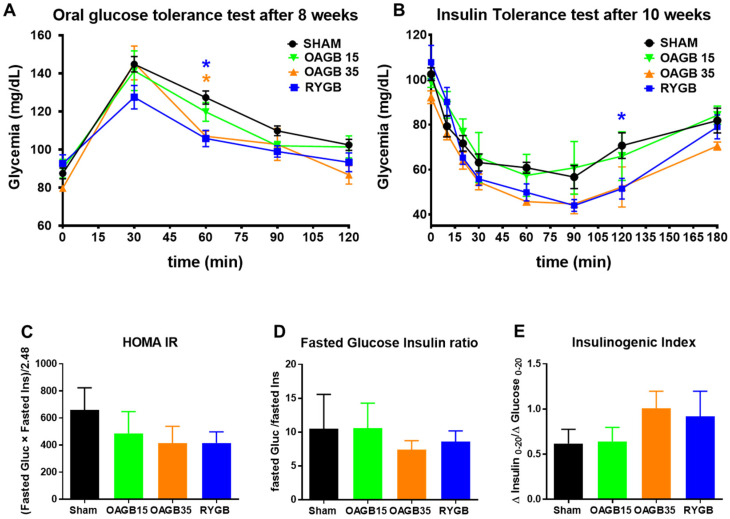
**Shortening the BPL of OAGB affects insulin sensitivity in obese rats**. (**A**,**B**) Glycaemia during an oral glucose tolerance test performed on week 8 post-surgery (**A**) or during an insulin tolerance test performed on week 10 post-surgery (**B**). * *p* < 0.05; for OAGB-35 (orange) or RYGB (blue) vs. Sham (black) by Dunnett’s multiple comparison tests after 2-way RM ANOVA. (**C**) Calculated HOMA-IR [(Fasted Glucose × Fasted Insulin)/2.48], (**D**) Fasted insulin over Fasted Glucose Ratio and (**E**) Insulinogenic index (Δ Insulin _0–20_/Δ Glucose _0–20_). No statistical difference was observed after a Kruskal–Wallis Test. Data are expressed as means ± SEM, SHAM n = 6, RYGB n = 6, OAGB15 n = 5, OAGB35 n = 4.

**Figure 4 jcm-11-04976-f004:**
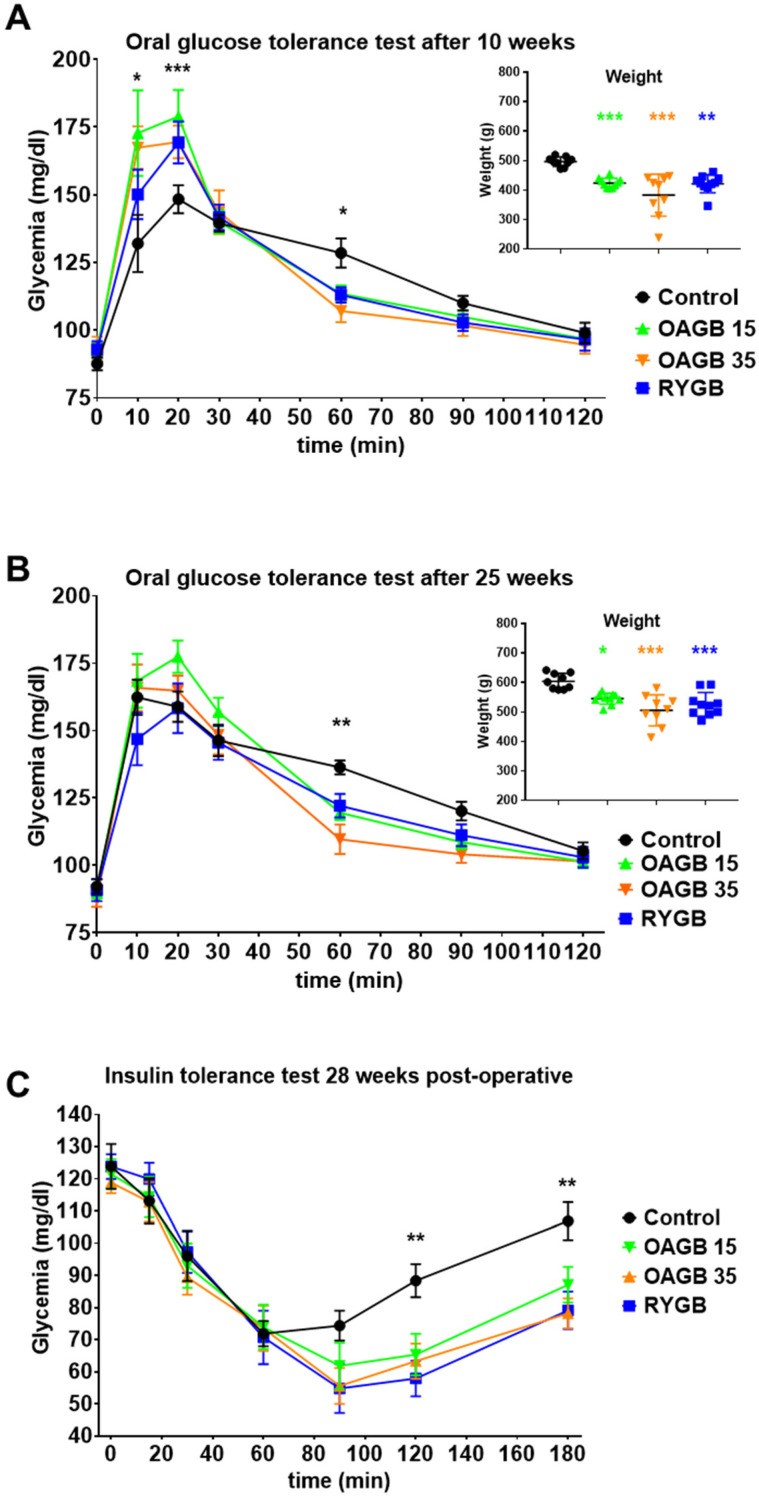
**Shortening the BPL of OAGB maintains amelioration of glucose control in non-obese rats**. (**A**,**B**) Glycaemia during an oral glucose tolerance test performed on week 10 (**A**) or 25 (**B**) post-surgery * *p* < 0.05 and ** *p* < 0.01, for control (black) animals versus animals operated on bariatric surgery by Dunnett’s multiple comparison tests after 2-way RM ANOVA. Inserts: corresponding weight of the animals * *p* < 0.05; ** *p* < 0.01, *** *p* < 0.001 for OAGB-15 (green) OAGB-35 (orange) or RYGB (blue) vs. Control (black) animals by Dunn’s multiple comparison tests after Kruskal–Wallis Test. (**C**) Glycaemia during an insulin tolerance test performed on week 28 post-surgery. ** *p* < 0.01 for control (black) animals versus animals operated on bariatric surgery by Dunnett’s multiple comparison tests after 2-way RM ANOVA. Data are expressed as means ± SEM, Control n = 8, RYGB n = 10, OAGB15 n = 11, OAGB35 n = 10.

## Data Availability

The data presented in this study are available on request from the corresponding author.

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
