# Peer review of "Shortening the Biliopancreatic Limb Length of One Anastomosis Gastric Bypass Maintains Glucose Homeostasis Improvement with Limited Weight Loss"

_jcm, 2022, doi:10.3390/jcm11174976_

Round 1
Reviewer 1 Report
This manuscript examines the effects of different gastric by pass procedures on weight loss, nutrient absorption and levels of circulating hormones involved in glucose and nutrient metabolism. The study is basically well designed and the manuscript is well written. However, the authors start the paper by saying gastric by pass surgery is the only way for people who are morbidly obese to lose weight. Could the authors please provide evidence that this is the case? There are also a number of details missing from methods and minor edits suggested below:
Line 70: Instead of performing you may want to say placing
Line 111 what does it mean when the authors say the stomach was tweaked with a staple gun?
Line 115: can the authors please describe the composition of the liquid diet? How were animals monitored post-surgery?
Line 208: may want to edit to say “animals undergoing bariatric surgery”
Author Response
We thank the 2 reviewers for the time they spent on our manuscript and the comment they made to ameliorate its quality and comprehension.
Please find below our point by point response to reviewer 1 comments, questions, and suggestions.
Reviewer 1
This manuscript examines the effects of different gastric by pass procedures on weight loss, nutrient absorption and levels of circulating hormones involved in glucose and nutrient metabolism. The study is basically well designed and the manuscript is well written. However, the authors start the paper by saying gastric by pass surgery is the only way for people who are morbidly obese to lose weight. Could the authors please provide evidence that this is the case? There are also a number of details missing from methods and minor edits suggested below:
We thank the reviewer for his/her positive comments on the quality of our manuscript.
We did not write that “gastric bypass surgery is the only way for people who are morbidly obese to lose weight”. We stated that “bariatric surgery (in general) is currently the only effective treatment for managing morbid obesity” we know precise that as “Although standard primary treatment of obesity consists of diet and lifestyle interventions, in absence of long-term success, bariatric surgery is still currently the only effective treatment for managing morbid obesity as it leads to rapid and long-lasting weight loss.” (lines 51-55)
Line 70: Instead of performing you may want to say placing
This is now corrected line 71
Line 111 what does it mean when the authors say the stomach was tweaked with a staple gun?
To perform the gastric pouch in OAGB or RYGB animals, the forestomach is resected using an Echelon 45-mm staple gun with blue cartridge (Ethicon) and then, the gastric pouch is created with a TA-DST 30-mm-3.5-mm stapler (Covidien). Thus, to mimic surgery in Sham animals the stomach is tweaked /pinched with a similar unarmed staple gun. For more details see references [12,13].
To clarify this point, we have now corrected the manuscript line 117-120: “For Sham rats, to mimic surgery, the stomach was pinched with an unarmed staple gun similar to the one used to perform the gastric pouch in RYGB and OAGB animals [12,13]”
Line 115: can the authors please describe the composition of the liquid diet? How were animals monitored post-surgery?
For clarification, we have now added the following information lines121- 129:
“For post-operative care, rats were kept in individual cages and monitored every 12 hours for 48 hours and then daily for weight, food intake, general behavior and wound healing with administration of analgesics and antibiotics (buprenorphine 0.05mg/kg, meloxicam 1mg/kg, penicillin 20,000 units/kg), if necessary. All rats (including control groups) were kept with no access to water and food during day 1 post-surgery but hydration was assured through injection of an isotonic polyionic solution (2 g KCl/L, 4g NaCl/L, 50g glucose/L). On day 2 and 3 post-surgery, rats had free access to water and a liquid highly digestible diet composed of 21% protein, 79% carbohydrate (C0200, Altromin), and on day 4, they had access ad libitum to a normal chow diet.
Line 208: may want to edit to say “animals undergoing bariatric surgery”
This has been corrected line 219.

Reviewer 2 Report
Thank you for allowing me to review the manuscript. Although a rat and not human study, I believe the authors have written the discussion in a way that does not overly extrapolate the clinical relevance and is appropriate for an animal surgical model.
I have one major concern. Is 4 per group really powered sufficiently to detect differences in outcomes of this study (particularly food intake, and calories lost, calories absorbed as well as the insulin sensitivity metrics)? Although this is brought up as a limitation, why could additional animals not have been added to ensure proper power? Please also report the individual group survival rate.
The methodology describes the studies around figure 4 as a non-surgical control. Please remove text in the results and figure legend which describe this as a sham surgery.
The results for Figure 4 describe the glycemic response as "independent of weight." However, the figure inset shows a significant difference in body weight. Without a weight matched group to the bariatric groups, the concept of weight independence can not be explored. This should be altered in the discussion as well.
Author Response
We thank the 2 reviewers for the time they spent on our manuscript and the comment they made to ameliorate its quality and comprehension.
Please find below our point by point response to Reviewer 2 comments, questions, and suggestions.
Reviewer 2
Thank you for allowing me to review the manuscript. Although a rat and not human study, I believe the authors have written the discussion in a way that does not overly extrapolate the clinical relevance and is appropriate for an animal surgical model.
We thank the reviewer for recognizing that we did not overstate the importance of our work and its relevance to human.
I have one major concern. Is 4 per group really powered sufficiently to detect differences in outcomes of this study (particularly food intake, and calories lost, calories absorbed as well as the insulin sensitivity metrics)? Although this is brought up as a limitation, why could additional animals not have been added to ensure proper power? Please also report the individual group survival rate.
We agree with the reviewer that the number of animals per group is low. The initial number was to be 6 per group as according to our previous studies, 5 animals per group were sufficiently powered to detect differences in weight loss and GLP-1 secretion with an alpha risk of 0.05 and a power 1-beta of 0.9. We lost 1 animal in the OAGB-15 group and 1 animal in the OAGB-35 group. In addition, at the time of the cull, we noticed that one OAGB 35 animal did not present a correct gastric pouch, this rat was thus excluded from the analyses shortening the number of animals to 4 only in this specific group. This set of experiment was done in the beginning of 2020 and it was impossible at that time to quickly set up new experiments due to COVID19 related lockdowns. This is why we are very careful in the interpretation of our results not overinterpreting the absence of statistical significance.
We are now a reporting the individual group survival rate :
Lines 97-101: “One OAGB-15 animal died 9 days after the surgery and one OAGB-35 animal died 1 day after the surgery. In addition, after the cull, we noticed that one OAGB-35 animal had a defective gastric pouch, this animal was thus excluded from the study. The final number of animals was n=6 for Sham, n=6 for RYGB, n=5 for OAGB-15, and n=4 for OAGB-35.”
Lines 105-107 : “One RYGB, one OAGB-15 and two OAGB-35 died during or shortly after the surgery. The final number of animals was n = 8 for control, n=10 for RYGB, n=11 for OAGB-15, and n=10 for OAGB-35.”
We also now precise the number of animals per group in each figure legend (lines 191-192, 206, 242-243 and 295).
The methodology describes the studies around figure 4 as a non-surgical control. Please remove text in the results and figure legend which describe this as a sham surgery.
This study is composed of 2 sets of experiments. In the 1st set with HFD fed animals, the control group underwent a sham surgery, whereas in the second set with ND fed animals, the control group was a non-surgical control. We try to be clear about this distinction even though in our experience, we did not observe any difference between sham and non-surgical group. Thus, in all the experiments but the one presented on figure 4, the sham group are really sham.
We thank the reviewer for pointing out our mislabeling concerning the studies around figure 4. We have now removed the term sham in the results and figure legend and corrected it as non-operated or control group (lines 268, 271, 291).
The results for Figure 4 describe the glycemic response as "independent of weight." However, the figure inset shows a significant difference in body weight. Without a weight matched group to the bariatric groups, the concept of weight independence can not be explored. This should be altered in the discussion as well.
The 2 figure insets show that there is only a difference in body weight between the operated group and the non-operated group. However, they are no statistical difference between the 3 operated groups thus we think it is fair to suggest that the glycemic response is not solely dependent on weight. We agree with the reviewer that the best way to address the concept of weight independence would be to have a weight matched group. The weight matched group is very difficult to obtain thus we have now claimed down our results and discussion appropriately (lines 277-278).

Round 2
Reviewer 2 Report
all comments and questions have been adequately addressed